# T cell responses in repeated controlled human schistosome infection compared to natural exposure

Emmanuella Driciru [1,2,3], Jan Pieter R. Koopman [1], Sanne Steenbergen[1], Friederike Sonnet[1], Koen A. Stam [1], Laura de Bes-Roeleveld[1], Eva Iliopoulou[1], Jacqueline J. Janse[1], Jeroen Sijtsma[1], Irene Nambuya[4], Stan T. Hilt [1], Marion König [1], Yvonne Kruize[1], Miriam Casacuberta-Partal[1], Moses Egesa[2,3,5], Govert J. van Dam [1], Paul L. A. M. Corstjens [6], Lisette van Lieshout[1], Harriet Mpairwe[2,7], Andrew S. MacDonald [4,8], Maria Yazdanbakhsh [1], Alison M. Elliott [2,7], Meta Roestenberg [1] & Emma L. Houlder [1] ✉

In *Schistosoma*-endemic regions a lack of natural sterilizing immunity means individuals are repeatedly infected, treated and reinfected. Due to difficulties in tracking natural infection, kinetics of host immune response during these reinfections have not been elucidated. Here, we use repeated (3x) controlled-human-*Schistosoma mansoni* infection (CHI) to study how antigen-specific T cells develop during reinfection (NCT05085470 study). We compared these responses to naturally infected endemic Ugandan individuals (HALLMARK study). A mixed Th1/Th2/regulatory CD4⁺ T cell response develops in repeated CHI. Adult-worm-specific responses after repeated CHI were similar to endemic-natural infection. However, endemic participants showed differential responses to egg- and cercariae-antigens. Repeated CHI with sequential exposure to cercariae of different sexes (male-female-male) revealed an elevated CD4⁺ T cell cytokine response to adult-worm and egg-antigens. Our findings demonstrate that single-sex schistosome infection elicits adult-worm-specific T cell cytokine responses that reflect endemic-natural infection. This study advances our understanding of the immunology of schistosome (re) infection in the human host.

Schistosomiasis is a neglected tropical disease caused by helminths of the *Schistosoma* species. Over 250 million people are affected globally, with the Sub-Saharan region accounting for more than 90% of infections[1]. Continuous infection with the long-lived *Schistosoma* parasites exerts a chronic immune modulatory effect on the host due

to prolonged exposure to schistosome antigens, primarily via prolific egg production[2]. Existing evidence on host immune response is mainly derived from murine studies, and shows a distinct immune trajectory across infection stages, with a T helper 1 (Th1) phenotype followed by Th2 and regulatory responses post egg production[3,4]. Egg-induced

[1]Leiden University Center for Infectious Diseases, Leiden University Medical Center, Leiden, Netherlands. [2]Schistosomiasis Focus Area, Vaccine Research Theme, Medical Research Council/Uganda Virus Research Institute and London School of Hygiene & Tropical Medicine Uganda Research Unit, Entebbe, Uganda. [3]Uganda Virus Research Institute, Entebbe, Uganda. [4]Lydia Becker Institute of Immunology and Inflammation, University of Manchester, Manchester, UK. [5]Department of Infection Biology, London School of Hygiene and Tropical Medicine, London, UK. [6]Department of Cell and Chemical Biology, Leiden University Medical Center, Leiden, the Netherlands. [7]Department of Clinical Research, London School of Hygiene and Tropical Medicine, London, UK. [8]Institute of Immunology and Infection Research, University of Edinburgh, Edinburgh, UK. ✉e-mail: e.l.houlder@lumc.nl

host immune responses are responsible for disease pathology and morbidity, although only 10% of cases progress to severe disease[5].

Reinfection due to repeated exposure, usually from early childhood, is common in endemic settings. Although praziquantel (PZQ) chemotherapy effectively reduces worm burden to undetectable levels, reinfection rapidly occurs. A recent study showed prevalence is restored to over 44.5% within 8 months of PZQ administration among shoreline communities of Lake Victoria, Uganda[6,7]. Over time, individuals in the highly endemic areas show reduced infection burden with age, after repeated infection-treatment cycles, attributable to partial immunity and/or altered exposure patterns[8,9]. Increased Th2 cytokines, high eosinophil count, and worm-specific IgE levels, have been associated with natural immunity, although not all are consistently found in meta-analyses[4,8,10–12]. How the immune response develops during natural repeated infection-treatment cycles, potentially leading to partial immunity, is still unclear. Natural infection studies are inherently limited by asynchronous and diverse exposure histories, hindering understanding of immune kinetics, which requires a precise definition of (re-) infection time points, dose, interval, and duration.

With the advent of controlled human infection (CHI) studies, it is now possible to probe in depth into infection immune dynamics using a precise longitudinal approach. In our previous single-sex single-infection *Schistosoma*-CHI model, Dutch participants showed symptom-associated inflammatory Th1 responses at 4 weeks post-infection, followed by Th2/regulatory responses at week 8[13]. Notably, Th2 response developed despite the absence of eggs, previously considered to be crucial for the switch to a Th2 response[14,15].

To replicate the natural schistosomiasis reinfection pattern in endemic settings, we designed a repeated *Schistosoma mansoni* (*Sm*) CHI model (repeated CHI), exposing *Sm*-naïve Dutch volunteers to three cycles of cercariae and PZQ treatment[16]. The previously published clinical arm of this trial investigated whether reinfection could induce protective immunity, and found that reinfection does not result in protective immunity, although symptoms decreased after the second and third exposure in the reinfection group[16]. During this trial, routine stool PCR and confirmatory microscopy after the third exposure unexpectedly detected a low number of *Sm* eggs in one participant. No eggs were detected in any other participants. A retrospective examination found that during the second exposure, five of twelve volunteers were unintentionally exposed to female cercariae[16]. These five volunteers received male-female-male (m-f-m) exposures at the first, second, and third time points, respectively, in contrast to the planned male-male-male (m-m-m) exposure (Fig. 1A). The possibility of egg production after the third exposure in the m-f-m exposed participants could have resulted from ineffective clearance of female worms by PZQ, allowing for worm pairing after the third (male) exposure[17]. Following the third exposure, m-f-m exposed volunteers (with potential eggs) tended to have higher eosinophil count, antibody, and serum cytokine levels[16].

In this exploratory immunological study, we investigated the specific cellular and cytokine response overtime during repeated CHI[16] and compared this to endemic-natural infection[18]. We aimed to, first, determine the cellular, and CD4 + T cell-specific cytokine response to *Sm* adult-, cercarial-, and egg- antigens during repeated CHI. Secondly,

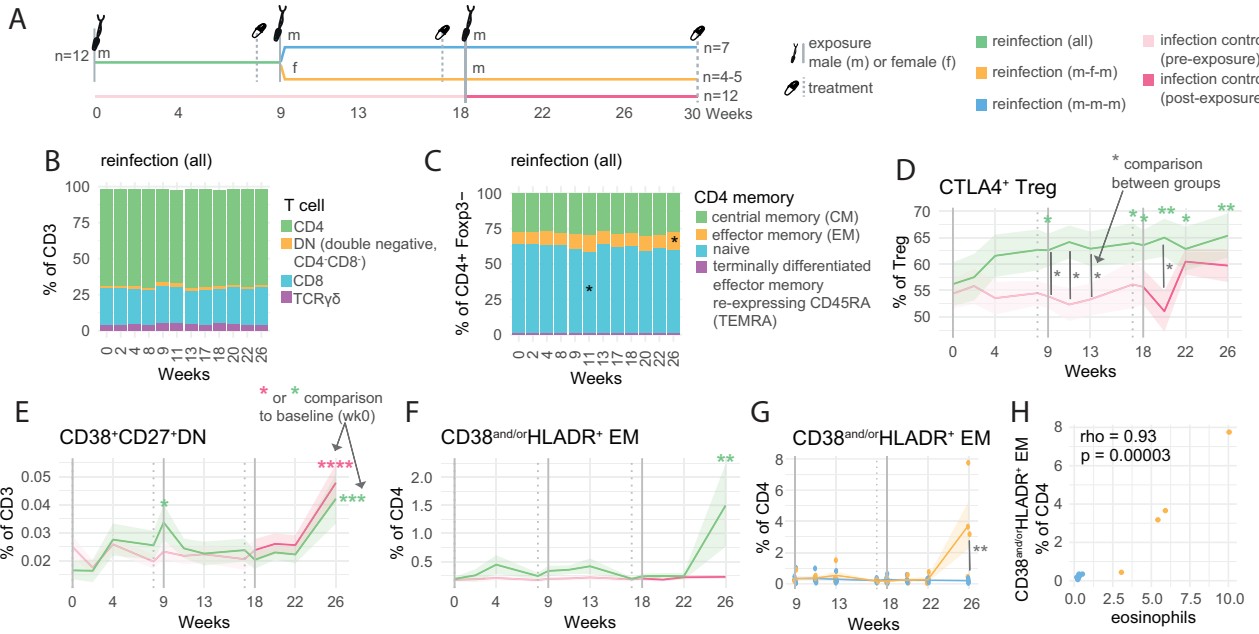

**Fig. 1 | Subtle alterations to T cell phenotype during repeat schistosome infection. A** Overview of trial design. Volunteers were randomised to reinfection (*n* = 12) or infection control (*n* = 12) groups. Reinfection volunteers (green) underwent three exposures (weeks 0, 9, and 18) and treatment (weeks 8, 17, and 30) cycles, whereas the infection control group (pink) underwent a single exposure (at week 18). At the second infection, *n* = 5 volunteers were unintentionally exposed to female cercariae (f) with groups referred to from then onwards as reinfection male-female-male (m-f-m, yellow) or reinfection male-male-male (m-m-m, blue). **B** Stacked bar chart of T cell subset frequencies. Each bar section represents the mean frequency. **C** Stacked bar chart of CD4 T cell memory frequencies. Each bar section represents the mean frequency. **D**–**F** Ribbon plots depicting mean (lines) frequency and standard error (shaded area) of the mean of specified populations, coloured by infection group. In panels (**B**–**F**), a two-sided linear mixed model was performed to (separately) assess changes in the reinfection and infection control

group compared to the week 0 baseline, with the volunteer as a random effect. False discovery rate (FDR) corrected *p* values are displayed in black in figure **C**, and green (reinfection) and pink (infection control) in figure (**D**–**F**). *FDR <0.05, **FDR <0.01, ***FDR <0.001, ****FDR <0.0001. To compare between reinfection (all) and infection control groups, Welch's *T*-tests were performed at each timepoint, with significant *p* values (uncorrected) displayed with a grey line *p < 0.05. **G** Ribbon plots depicting mean (lines) frequency and standard error (shaded area) of the mean of CD38and/or HLADR+ effector memory (EM) CD4 T cells in the reinfection (m-m-m) and reinfection (m-f-m) groups. To compare between reinfection (m-m-m) and reinfection (m-f-m) groups, two-sided Mann–Whitney tests were performed at each timepoint with significant *p* values (uncorrected) displayed with a grey line *p < 0.05, **p < 0.01. **H** Scatter plots comparing CD38 and/or HLADR+ EM CD4 T cell frequency to blood eosinophilia at week 26. Spearman's rank correlation analysis was performed with the correlation coefficient rho and *p* value reported.

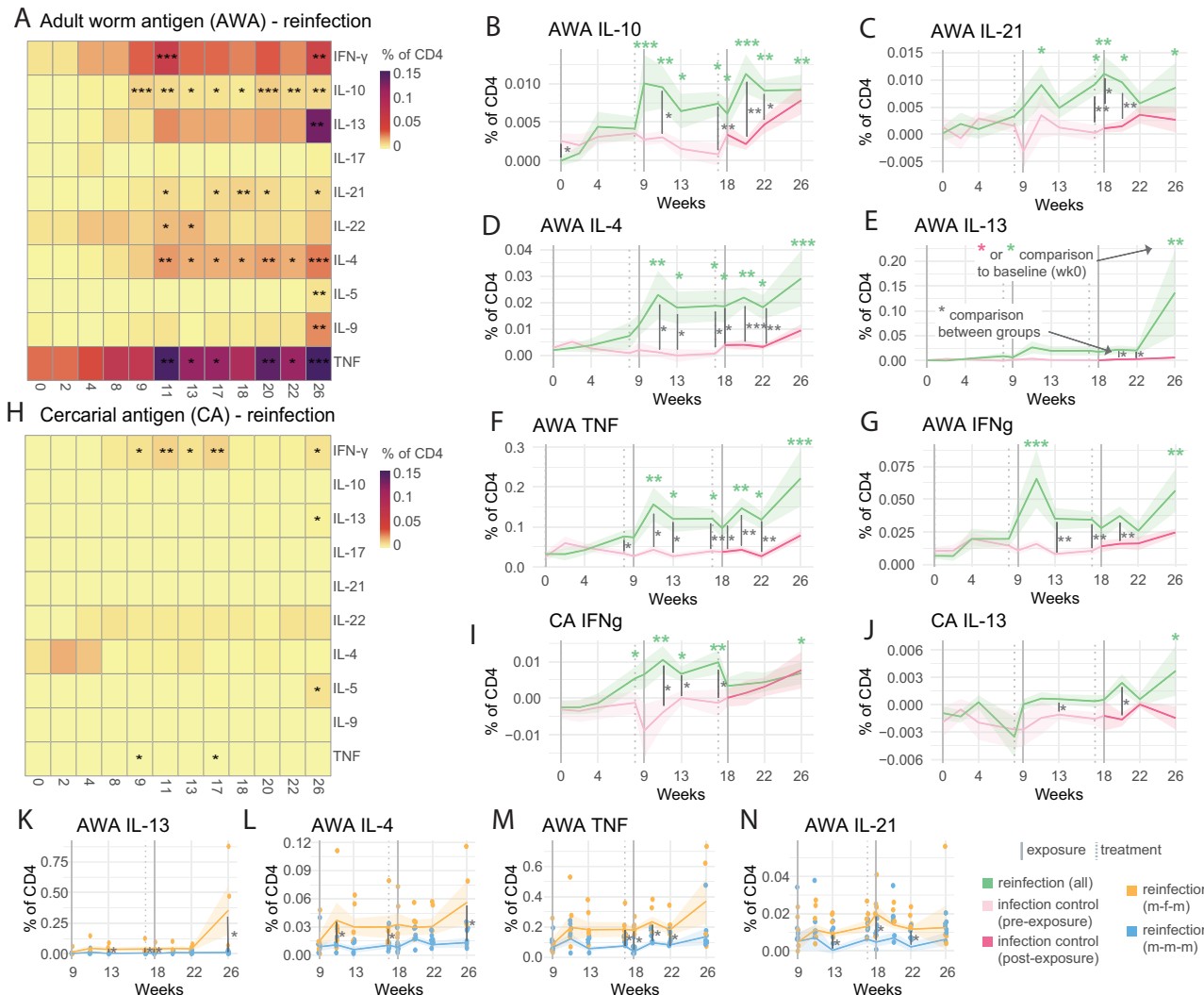

**Fig. 2 | Robust worm (AWA) but not cercariae (CA) - specific CD4 T cell responses during repeat schistosome infection. A** Heatmap displaying frequencies of cytokine-positive CD4+ T cells post AWA stimulation of PBMCs from reinfection volunteers, coloured by frequency. **B–G** Ribbon plots depicting mean (lines) frequency and standard error (shaded area) of the mean of cytokine production in AWA-stimulated CD4 T cells, coloured by infection group, comparing reinfection (all) and infection control. **H** Heatmap displaying frequencies of cytokine-positive CD4+ T cells post CA-stimulation of PBMCs from reinfection volunteers, coloured by frequency. **I, J** Ribbon plots depicting mean (lines) frequency and standard error (shaded area) of the mean of cytokine production in CA-stimulated CD4 T cells, coloured by infection group, comparing reinfection (all) and infection control. In panels (**A–J**), a two-sided linear mixed model was performed to (separately) assess changes in the reinfection and infection control group compared to the week 0 baseline, with the volunteer as a random effect. FDR values are displayed in black (panels **A**, **B**) or green (panels **C–J**) for reinfection (all) or red (infection control) *FDR <0.05, **FDR <0.01, ***FDR <0.001. To compare between reinfection (all, n = 11–12) and infection control groups (n = 11) two-sided Welch's *T*-tests were performed at each timepoint with significant *p* values (uncorrected) displayed with a grey line *p < 0.05, **p < 0.01. **p < 0.001. **K–N**. Ribbon plots depicting mean (lines) frequency and standard error (shaded area) of the mean of specified stimuli and populations, coloured by infection group, comparing reinfection (m·f·m) and reinfection (m·m·m). To compare between reinfection (m·m·m, n = 7) and reinfection (m·f·m, n = 4–5) groups, two-sided Mann–Whitney tests were performed at each timepoint with significant *p* values (uncorrected) displayed with a grey line *p < 0.05. All frequencies shown are post medium subtraction.

to compare the *Sm*-specific T cell profile, and CD4 + T cell cytokine response to *Schistosoma* life-stage-specific antigens during repeat CHI to that in endemic-natural infection. We hypothesized that experimental *Schistosoma*-reinfection exerts an immune modulatory response in the human host similar to that in natural-endemic settings, with a possible immune-protective impact. Using spectral flow cytometry to analyze cellular responses in peripheral blood mononuclear cell (PBMC) samples from the repeated CHI trial, we observed a mixed Th1/Th2/regulatory response, primarily directed against adult worm antigens (AWA). T cell response in repeated CHI was comparable in magnitude to that in endemic-natural infection but lower than in the repeated CHI volunteers that were sequentially exposed to schistosome cercariae of different sexes (m·f·m exposure).

## Results

### Single-sex repeated infection challenge alters the CD4 + T cell phenotype

To investigate cellular changes in the T cell compartment induced by repeated CHI (Fig. 1A), PBMC samples were analyzed using a 30+ marker spectral flow cytometry panel. At week 11 (2 weeks post second exposure), there was a significant reduction in the frequency of naïve CD4 + T cells in the reinfection (all) group compared to baseline (week 0) (Fig. 1C). Such a decrease may indicate the possible onset of adaptive responses following PZQ-induced worm antigen unmasking[19–21]. Moreover, at week 26, effector memory (EM) T cells increased in the reinfection (all) group (Fig. 1C). Subsets in the total CD4+ T cell compartment (CD4+ T cells, CD8+ T cells, γδT cells or DN (double negative

CD4−CD8−) T cells) remained unchanged (Fig. 1B). In the infection control group that received a single exposure at week 18, no significant alterations in any T cell populations were observed (Supp. Fig. 1A, B).

Next, we examined changes in the CD4 T cell activation status. By week 9, CTLA4+ regulatory T cells increased significantly in the reinfection (all) group compared to week 0 baseline, staying elevated throughout the second and third infections (Fig. 1D). Moreso, at weeks 9, 11, 13, and 22 this regulatory T cell subset was significantly higher in the infection group compared to infection control group which did not show any significant increases (Fig. 1D). Similarly, at week 9 and 26 a CD38+CD27+ DN T cells increased significantly in the reinfection (all) group, as well as week 26 in the infection control group (Fig. 1e). This population is thought to be regulatory, and has been previously shown to increase after single-exposure[13].

At week 26 (8 weeks post third exposure), activated CD38AND/ORHLADR+ EM T cells significantly expanded in the reinfection (all) group (Fig. 1F)[13]. Other CD4 T cell populations−CD4+ Treg, PD1+ Treg, CD38/HLADR+ Treg, Gata3+CRTH2+ CD4+ T cell, CTLA4+CD4+ T reg, and PD1+CD4+ T cells−remained generally stable (Supp. Fig. 1C–H). When m·f·m and m·m·m reinfection groups were split, we observed that the expansion of CD38AND/ORHLADR+ EM T cells at week 26 was attributable to m·f·m volunteers, with potential egg production (Fig. 1G). Significant differences between the m·f·m and m·m·m groups were not observed in other CD4 T cell populations (Supp. Fig. 1I–P). Notably, CD38AND/ORHLADR+ EM T cell expansion significantly correlated with peripheral eosinophilia[16], suggestive of a concerted inflammatory/type-2 response in these m·f·m volunteers (Fig. 1H).

Overall, these results demonstrate modest increases in the regulatory cell (CTLA4+ Treg and DN T cell) populations in single-sex (m·m·m) reinfection. Meanwhile, in m·f·m volunteers, a striking increase in activated CD38AND/ORHLADR+ EM T cells was observed at week 26, following potential egg production.

## Repeated infection induces a mixed CD4+ T cell-specific cytokine response to adult worm antigen, but not cercarial antigen

To understand if repeated infection changes T cell cytokine response to the different schistosome life cycle stages, PBMCs were stimulated with adult worm (AWA) and cercarial (CA) antigens. A robust CD4+ T cell cytokine response was observed upon AWA stimulation in the reinfection (all) group. This response comprised of significant elevation in Th1, Th2, and regulatory cytokines−TNF, IFN-y, IL-13, IL-4, IL-9, IL-5, IL-10, IL-22, and IL-21 - compared to the baseline (Fig. 2A–G and Supp. Fig. 2A–E). AWA-specific TNF and IL-22 were significantly increased in the reinfection group over infection-controls during the first infection (wk 4−8) (Fig. 2F and Supp. Fig. 2C). Generally, however, AWA-specific responses were most visible at week 11, after the first PZQ administration (Fig. 2A–G). This could be attributed to immune boosting by the second exposure (2 weeks prior) and or antigen release due to PZQ treatment (3 weeks prior)[22]. In contrast, a muted response to CA was observed in the reinfection (all) group, with significant increases occurring only in IFN-y, TNF, IL-5, and IL-13 cytokine levels (Fig. 2H–J, Supp. Fig. 2F–N). As expected, changes in antigen-specific cytokine response in the infection control group were observed only after cercariae exposure at the third exposure timepoint (Supp. Fig. 2A, F).

Within the reinfection group, the m·f·m group had significantly higher AWA-specific IL-13, IL-4, TNF, IL-2, and IFN-y than the m·m·m group from week 13, but most strikingly at week 26 (Fig. 2K−N and Supp. Fig. 2O−X). Cytokine changes were not entirely driven by the m·f·m group. AWA-specific cytokines increased in the m·m·m group, compared to baseline week 0 (IL-4, IL-13, IL-10, IFN-γ, and TNF) and infection control (IL-4, IL-10, IFN-γ, and TNF) (Supp. Fig. 3A−E).

Taken together, repeated CHI elicits a robust and mixed−Th1/Th2/regulatory CD4+ T cell-specific cytokine response to AWA, and a limited response to cercarial antigen. Schistosome-specific CD4 T cell

responses tend to increase after the first infection/treatment cycle (week 11 onwards) and are notably higher in the m·f·m group at week 26.

## Enhanced inflammation due to m·f·m exposure is short-lived, tending to reduce by week 30

Having observed robust alterations in the CD4+T cell compartment in the m·f·m group at week 26, we asked if this response would continue to increase at week 30 (the latest pre-treatment time point) (Fig. 3). We observed that CD38AND/ORHLADR+ EM T cells increased at week 26 (Fig. 1G, 3A), then decreased by week 30 in the m·f·m group, while remaining significantly higher than in the m·m·m group (Fig. 3A). A similar pattern was observed with serum cytokines TNF, CCL23, and CCL4, which tended to decrease by week 30 in the m·f·m group (Fig. 3B). No significant changes in IL-18 or CXCL10 were observed (Supp. Fig. 4A).

Stimulations with adult worm (AWA), egg (SEA), and cercarial (CA) antigens were performed, to understand how schistosome stage-specific responses changed between weeks 26 and 30. We observed a robust SEA, AWA (but not CA) responses at week 26 in the m·f·m p, higher than in the m·m·m group, and encompassing a wide range of Th1/Th2/regulatory cytokines (Fig. 3C−E, Supp. Fig. 4B−D). Cytokine responses to AWA tended to decrease by week 30 (except IL-4) (Fig. 3C). In comparison, and with the notable exception of IFN-y, responses to SEA tended to remain stable to week 30 (Fig. 3D).

## Muted responses in endemic-natural infection are more comparable with m·m·m than m·f·m repeated CHI participants

Next, we explored how T cell and cytokine responses in repeated CHI (m·m·m and m·f·m groups)[16] compare to that in natural-natural (mixed-sex) infections[18]. Repeated CHI samples were selected at the last time point before the final PZQ treatment (week 30) when their exposure history most closely echoed endemic populations. Endemic-infected individuals were chosen from a population with regular lake contact (entailing repeated exposure), varying infection burdens, and exposure histories (Supp. Fig. 5). These samples were grouped into "infected" and "uninfected" groups, based on stool Kato-Katz, supported by serum-circulating anodic antigen (CAA) and urine-circulating cathodic antigien (CCA) assays (Supp. Fig. 5)[18]. To note, endemic samples had reduced viability−with median viability of 70−80% for endemic participants −compared to 96% in repeated CHI (all) group (Supp. Fig. 6A). However, we found no evidence that viability was correlated to T cell phenotype or cytokine expression within endemic samples (Supp. Fig. 6).

Using PCA to define an overview of how unstimulated T cell phenotypes differed between m·m·m, m·f·m, and natural-endemic-infection groups, we observed that the groups tended to separate along the principal components (Fig. 4A). Activated CD38AND/ORHLADR+ EM T cells were highest in the m·f·m repeated CHI group (Fig. 4B). Endemic-infected individuals tended to have higher frequencies of co-inhibitory ligand CTLA4 as well as PD1 expressing CD4 T cells though unexpectedly lower Foxp3+CD25+ Tregs when compared to repeated CHI groups (Fig. 4B). Meanwhile, serum cytokines in endemic-infected individuals tended to most closely resemble m·m·m reinfection, although with increased regulatory cytokine IL-10 levels and reduced eosinophil-associated chemokine CCL23 levels (Fig. 4C). The m·f·m group had higher serum levels of TNF, CCL23, and CCL4, an effect mainly driven by two of the participants with exaggerated responses in all outputs (Fig. 4C). Notably, endemic serum samples had an overall more anti-inflammatory state, with increased IL-10 to TNF ratio (Fig. 4D).

We also compared the observed responses in endemic-infected and endemic-uninfected individuals. No significant differences in T cell phenotypes were observed in infected and non-infected participants, potentially attributable to heterogeneity in these populations

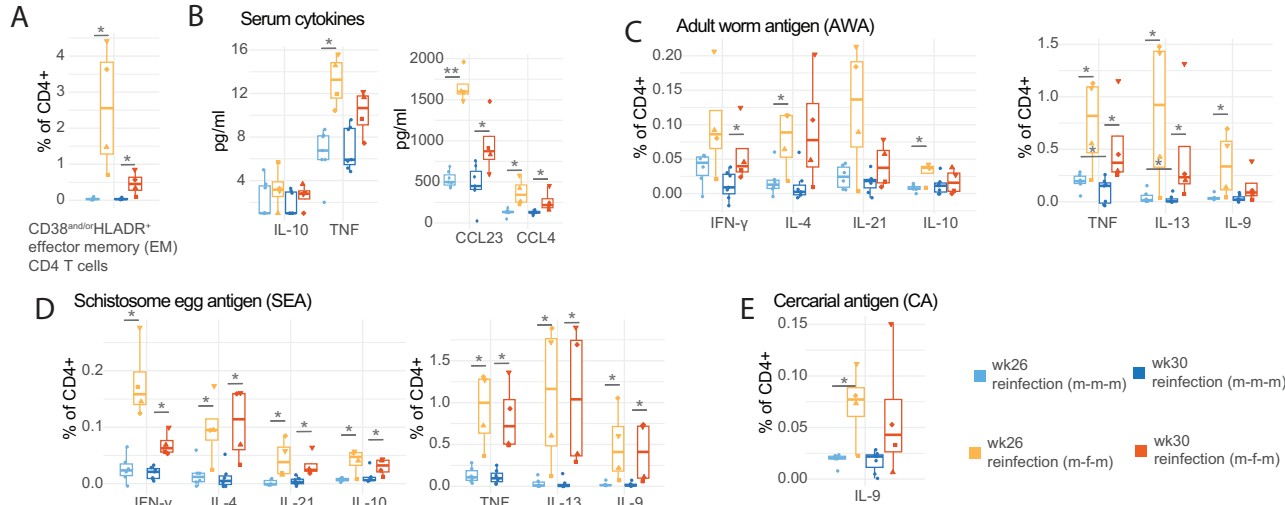

**Fig. 3 | Enhanced inflammation at week 26 in the reinfection (m-f-m) group tends to reduce by week 30. A** Boxplot comparing CD38^and/or^HLADR⁺ EM CD4 T cell frequency between reinfection (m-m-m) and reinfection (m-f-m) at week 26 and 30. **B** Boxplot comparing serum cytokines between reinfection (m-m-m) and reinfection (m-f-m) at weeks 26 and 30. **C–E** Boxplots comparing AWA-specific (**C**), SEA-specific (**D**), and CA-specific (**E**) CD4 T cell cytokine frequency between reinfection (m-m-m) and reinfection (m-f-m) at weeks 26 and 30. All frequencies shown are post medium subtraction. Differing shapes in reinfection (m-f-m) volunteers denote different individuals and are consistent between panels. Sample sizes are n = 6 for m-m-m at week 26, n = 7 at week 30, and n = 4 for m-f-m at both time points. Boxplots display the central line (median), and hinges (25th and 75th percentile) with whiskers extending from the hinge to the largest/smallest value or 1.5x the interquartile range. P values (uncorrected) are displayed, derived from two-sided Wilcoxon signed-rank tests when comparing between time points, or two-sided Mann–Whitney test to compare reinfection (m-m-m) and reinfection (m-f-m) groups. For all comparisons: \*p < 0.05, \*\*p < 0.01, \*\*\*p < 0.001.

(Fig. 4E–F). However, serum cytokines CXCL10 and IL-10 were significantly elevated in the infected endemic individuals compared to their uninfected counterparts (Fig. 4G). This increase in CXCL10 has been previously reported in this sample set[18]; however, our study extends these findings by demonstrating that the CXCL10 level observed is comparable to that seen after repeated CHI (Fig. 4C)[13,17,23]. No differences in TNF, CCL23, or CCL4 serum levels were observed (Fig. 4G). In endemic samples, schistosome infection leads to a more anti-inflammatory state, with an increased IL-10 to TNF ratio (Fig. 4H).

Next, we compared schistosome stage-specific responses in the repeated CHI and endemic infection groups. PCA of individual samples post-stimulation revealed a distinct clustering of the SEA-stimulated m-f-m group, seen to a lesser extent in AWA stimulation and not observed following CA-stimulation (Fig. 5A, D, G). Considering specific CD4⁺ T cell cytokines, SEA and AWA antigens elicited the highest response, with CA-specific responses the least. AWA and SEA-specific Th1/Th2/regulatory cytokine levels were highest in the m-f-m group, significant for SEA-specific TNF, IL-13, IL-9, IFN-γ, IL-4, IL-21, IL-10, and AWA-specific TNF, IL-13 and IFN-γ (Fig. 5B, E). Interestingly, single-sex repeated CHI (m-m-m) and natural-endemic infection elicited comparable antigen-specific cytokine responses, with a notable exception of IL-17, which was higher in endemic infection (Fig. 5B, E, H). When comparing endemic-infected and uninfected participants, specific responses were observed to AWA and CA (Fig. 5C, I), with close to an absent response to SEA (Fig. 5F). Infected endemic groups exhibited significantly higher levels of AWA-specific IL-4 and IL-10 and CA-specific TNF, IL-9, and IL-17 in comparison to uninfected endemic subjects (Fig. 5C, I). No significant changes in IL-5 and IL-22 were observed (Supp. Fig. 4F–H).

Taken together, the immune response in endemic infection is most comparable to that elicited after m-m-m repeated CHI, especially to adult worm antigen. No responses to SEA were observed in endemic participants, while the CA-specific Th17 response was elevated. Participants in the m-f-m group, with potential egg exposure, showed the highest cytokine response to both AWA and SEA.

## Discussion

This study has delineated how T cell and cytokine responses develop during repeated *S. mansoni* reinfection and treatment cycles; a study design that resembles exposure patterns seen in schistosomiasis-endemic areas. *Schistosoma*-reinfection is characterized by expanded CTLA4⁺ Tregs, CD38⁺CD27⁺DN T cells, and, in the m-f-m group, CD38^AND/OR^HLDR⁺ effector memory expansion. An increase in schistosome worm-specific CD4⁺ T cell cytokine levels consisting of a mixed Th1/Th2/Treg cytokine profile was seen in m-m-m participants. Before final treatment (week 30), AWA-specific responses observed in the m-m-m group were of a similar magnitude to those in endemic-infected participants. Notably, endemic-infected individuals showed an anti-inflammatory profile, with a negligible response to SEA and an elevated Th17 response to CA. Reinfection m-f-m participants had the highest cytokine responses at weeks 26–30, possibly reflective of an acute response to egg-associated[3,14,15] and or female worm-specific antigens[24].

Changes in T cell phenotype during m-m-m reinfection were characterized by increased CTLA4⁺ Tregs, potentially contributing to a downregulated host-immune activation, increased parasite survival[25–27] and the reduced symptoms as observed during reinfection[16]. Treg expansion in endemic-natural infection has previously been described[28]. In addition, another potentially regulatory T cell subset, CD38⁺CD27⁺DN T cells, expanded at week 8–9, after the first infection, as in the previous single-sex 1x CHI[13]. Besides changes in the regulatory T cell phenotype, we also observed a decline in naïve T cells at 11 weeks post-exposure. A declining naïve population post-PZQ treatment could be attributed to immune maturation after antigen release from dying worms[19–21].

In terms of cytokine response, CD4⁺ T cell-specific cytokine responses in the m-m-m group were elevated during the first infection cycle, becoming significant after the second exposure. Interestingly, this cytokine response comprised a mixed array of Th1/Th2/regulatory cytokines; IFN-γ, TNF, IL-4, IL-13, IL-21, and IL-10, contrary to the long-held view of polarized initial Th1 responses that switch to Th2 on egg

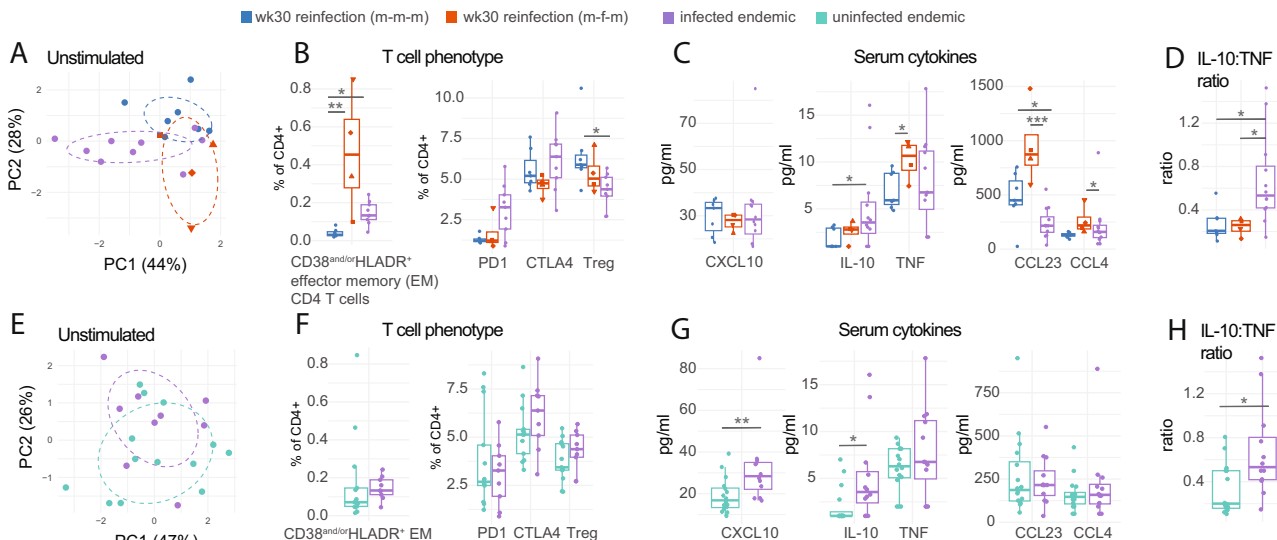

**Fig. 4 | Comparing T cell phenotype and serum cytokines in controlled human infection and endemic infection. A–D** Comparisons of week 30 reinfection male-male-male (m·m·m, $n = 7$), reinfection male-female-male (m·f·m, $n = 4$) and endemic schistosome infected ($n = 9$–12) and uninfected individuals ($n = 13$–16). Samples are coloured by group. **A** Principal component analysis (PCA) was applied to individual samples on the basis of frequencies of T cell populations. **B** Boxplots comparing T cell phenotype. **C** Boxplots comparing serum cytokine levels. **D** Boxplot comparing serum IL-10:TNF ratio. **E–H** Comparisons of schistosome uninfected and infected endemic ($n = 9$) individuals. Samples are coloured by group. **E** PCA applied to individual samples on the basis of frequencies of T cell populations. **F** Boxplots comparing T cell phenotype. **G** Boxplots comparing serum cytokine levels. H) Boxplot comparing serum IL-10:TNF ratio. Boxplots display the central line (median), and hinges (25th and 75th percentiles) with whiskers extending from the hinge to the largest/smallest value or 1.5x the interquartile range. *P* values are displayed, these are derived from two-sided Dunn's test for pairwise multiple comparisons in (**B–D**) and from a two-sided Mann–Whitney test in (**F–H**). \**p* < 0.05, \*\**p* < 0.01, \*\*\**p* < 0.001.

production[29,30]. We also noted that a significant magnitude of cytokine response was directed to AWA in m·m·m reinfection and a contrastingly negligible CA response. This finding may support the notion that viable cercariae, such as those used in CHIs, rapidly transverse the skin tissue and induce regulatory factors ensuring minimal host immune priming[31–33].

Compared to the m·m·m group, m·f·m volunteers showed elevated T cell responses, particularly from week 26. Due to reduced PZQ-sensitivity of female worms[17,34], male-female pairing may have occurred in the m·f·m volunteers following the third exposure. Immune differences between the m·m·m and m·f·m groups appeared from week 26, a time point that coincides with potential egg production[35]. Elevated Th2 and Th1 responses to egg and worm antigens at this timepoint coincide with male-female worm pairing and therefore could suggest an acute response to potential egg presence, a well-characterized phenomenon in the murine model[14]. Activated CD38[AND/OR]HLA-DR+ EM T cells were significantly increased in the m·f·m reinfection group. CD38 and HLA-DR are markers of activation and proliferation expressed by recently activated memory T cells and are often identified in acute inflammatory responses[36–38]. In this study, the observed T cell activation underscored elevated pro-inflammatory serum chemokines and a cytokine response consisting of CCL23, CCL4, and TNF, further supporting a proposed acute inflammatory response to egg antigens.

However, egg presence was not confirmed by egg microscopy or PCR in most (4/5) of these volunteers, potentially due to insufficient sensitivity to detect low levels of eggs. The observed response may stem from factors beyond detectable egg output, notably, the response to female worm-specific antigens. Murine studies have shown differential immune responses in female compared to male infection, and female worms express distinct proteins and glycosylation patterns, which may modify host immune response even in the absence of patent egg production[24,39,40]. However, in our previous single-sex CHI models, the immune profile and magnitude in the female-only was comparable to that in the male-only CHI model[13], suggesting female-specific factors alone may not sufficiently explain the m·f·m reinfection response. Alternatively, proteins expressed during pairing, mating, and or early stages of oogenesis following potential male-female worm interaction could underlie the observed response[24,41,42].

Next, we explored if repeated CHI in non-endemic (Dutch) participants induced immune responses reflective of naturally infected endemic (Ugandan) populations. Naturally infected Ugandan participants showed reduced inflammation, with a tendency for higher co-inhibitory markers (PD1+ and CTLA4+) and a regulatory serum cytokine environment (elevated IL-10:TNF ratio) compared to repeated CHI participants. This anti-inflammatory phenotype in endemic areas may be attributable to several factors (diet, pathogen exposure, microbiota), and has been associated with a dampened vaccine response[39,40,43]. In line with this, endemic-infected participants showed a negligible egg-specific response, a finding in line with previous studies[30,41,44,45]. This dampened egg response may serve to limit immunopathology arising from the host's fibrogenic reaction to deposited eggs[46,47]. AWA-specific responses in the endemic-infected participants were of comparable magnitude to reinfection (m·m·m) CHI volunteers at week 30, though less than that seen in reinfection (m·f·m) volunteers. Finally, an IL-17 response to CA was elevated in endemic-infected participants, potentially representing a protective barrier response, as this cytokine has been linked with protection against infection in murine models[48]. A limitation of this study is that we could not comprehensively screen participants for prior and current (co-)infections, with exposure histories likely differing in endemic and CHI volunteers, potentially contributing to observed differences. For example, over 95% of 5-year-old children in the endemic area of Entebbe, Uganda has experienced cytomegalovirus, and 60% malaria infection by age 5[49]. Malaria is not present in the Netherlands, and the lifetime seroprevalence of cytomegalovirus in the Netherlands is only 45%[50].

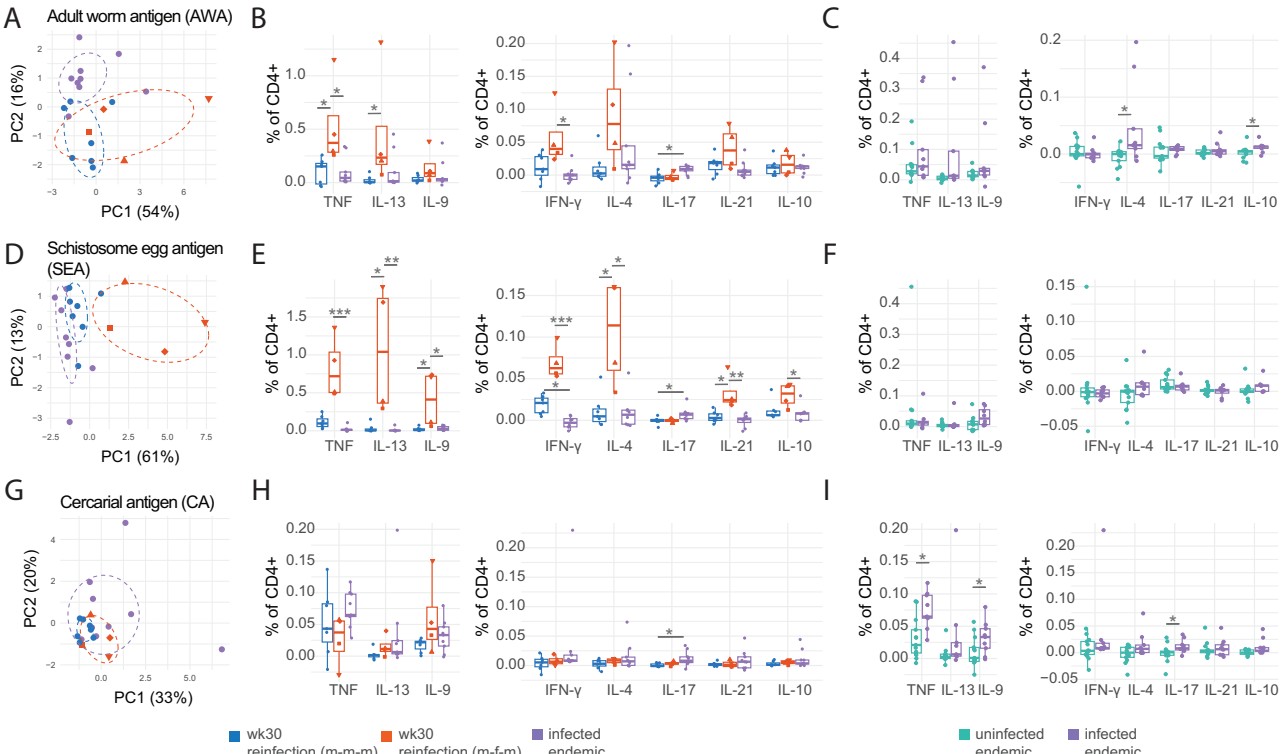

**Fig. 5 | Responses to different schistosome stages (AWA, SEA, CA) in controlled human infection and endemic infection. A, B, D, E, G, H** Comparisons of reinfection male-male-male (m-m-m, $n = 6$), reinfection male-female-male (m-f-m, $n = 4$), and endemic schistosome-infected ($n = 9$). Samples are coloured by group. **C, F, I** Comparisons of uninfected ($n = 13$) and schistosome-infected ($n = 9$) endemic individuals. Samples are coloured by group. **A** PCA applied to individual samples based on AWA-specific cytokine frequencies. **B, C** Boxplots comparing AWA-specific CD4 T cell cytokine frequencies. **D** PCA applied to individual samples based on SEA-specific cytokine frequencies. **E, F** Boxplots comparing SEA-specific CD4 T cell cytokine frequencies. **G** PCA applied to individual samples based on CA-specific cytokine frequencies. **H, I** Boxplots comparing CA-specific CD4 T cell cytokine frequencies. All boxplots display the central line (median), and hinges (25th and 75th percentile) with whiskers extending from the hinge to the largest/smallest value or 1.5x the interquartile range. All frequencies shown are post medium subtraction. *P* values are displayed, these are derived from two-sided Dunn's test for pairwise multiple comparisons in (**B, E, H**), and from a two-sided Mann–Whitney test in (**C, F, I**). *$p < 0.05$, **$p < 0.01$, ***$p < 0.001$.

Utilizing advanced cytometry techniques and a CHI model with the inherent advantage of a defined schistosome cercariae infection dose, sex, and timing has enabled us to conduct a longitudinal assessment of immune changes to reinfection over time. No volunteers developed resistance to infection[16], suggesting that the mixed Th1/Th2 response we observed is not sufficient for protection and that vaccines should aim to induce a different T cell phenotype, magnitude, or specificity. Our findings are in line with the "happy valley hypothesis", which argues that whilst extreme Th1 or Th2 responses can be protective, the balanced Th1/Th2 response we observe here may allow for worm survival[51]. It is possible that developing protective immunity may require years of exposure/reinfection and treatment cycles, with the shorter duration of CHI models presenting a potential limitation. Moreover, CHI models are limited by their incapacity to replicate the complex natural immunological and environmental context. Challenge strains often do not represent circulating wild-types, and CHI volunteers, such as in this repeated CHI, may differ from vaccine target populations in terms of comorbidities, microbiome, immunogenetics, and prior-exposure history[52]. A limitation of this repeated CHI model is that the initial design was intended for a single-sex, male-only reinfection. Hence, whether male-female worm interaction (and egg production) occurred in the unintended m-f-m group remains uncertain, given that stool PCR was negative for 4/5 participants.

Despite these limitations, the CHI model provides a precise alternative to natural infection studies where heterogenous pre-exposure levels, undefined infection timing, and inconsistent infectious doses limit definitive investigations into infection and disease prognosis. The controlled standardized infection conditions in CHI allow for detailed longitudinal studies of pathogen-host interactions, with a small number of volunteers[52]. Using this system, we have demonstrated how immune responses can develop during repeated *S. mansoni* infection/treatment cycles. Beyond progressing scientific understanding, CHI models can provide proof-of-concept for vaccine efficacy[53]. Understanding the differences in immune responses between non-endemic CHI and endemic infection is critical to infer how translatable vaccine studies performed in non-endemic CHI are to the endemic situation. Here, we demonstrate that worm-specific responses after repeated m-m-m CHI immune broadly resemble those of naturally infected endemic participants. However, immunological differences remain between the two groups, as demonstrated by non-overlapping PCA clustering, including increased IL-17 levels and reduced Th1 (IFN-γ and TNF) responses in endemic-infected volunteers compared to m-m-m repeated CHI. Therefore, transferring the CHI model to an endemic area is a critical next step to gain a holistic insight into how prior-exposure, co-infections, immune regulation, and repeated treatment may influence host responses to *Sm* (re)infection and vaccine-candidates[43] using the controlled longitudinal CHI design.

## Methods
### Study design
This research complies with all relevant ethical regulations. This study utilized samples from a repeated controlled human challenge infection (CHI) model (NCT05085470), a placebo-controlled randomized trial conducted at the Leiden University Medical Centre (LUMC), the

Netherlands[16], and the endemic HALLMARK study was conducted at the Uganda Virus Research Institute (UVRI), Entebbe, Uganda[18]. The endemic HALLMARK study was reviewed and approved by the UVRI Research Ethics Committee, as well as the Uganda National Council for Science and Technology and the University of Manchester Research Ethics Committee. For the repeated CHI trial, ethics approval was obtained from the local ethics review committee (Medisch-Ethische Toetsingscommissie Leiden Den Haag Delft [METC LDD], P21.070). Prior to any study procedure, informed consent was obtained from all participants.

In the repeated CHI trial, participants in the reinfection group ($n = 12$) were exposed to 20 cercariae at three time points; weeks 0, 9, and 18), with one group receiving male-only cercariae at all time points (m-m-m group, $n = 7$) and the other ($n = 5$) received a sequential exposure to cercariae of different sexes (m-f-m) at the three exposure time points, respectively. One participant in the reinfection m-f-m group was lost to follow-up shortly after the third exposure and was given PZQ treatment to clear the infection. The infection control group ($n = 12$) received two mock challenges at weeks 0 and 9, and one 20 male-only cercariae dose exposure at week 18. Praziquantel (PZQ) treatment was administered 8 weeks following the first and second exposures (weeks 8 and 17) and at 12 weeks after the third challenge (week 30) with 60 mg/kg PZQ[16]. At weeks 8 and 17, the infection control group was given a placebo treatment in place of praziquantel. *Schistosoma*-specific PCR was performed on stool samples, as previously described[54] to detect the presence of schistosome eggs. Endemic human samples were obtained from 18-to 25-year-old young adults living in the schistosomiasis-endemic area of Kigungu landing site in Entebbe municipality, Uganda[18]. This was a cross-sectional study consisting of volunteers with no active infection at the point of enrolment (endemic-uninfected group, $n = 16$) and those with active infection (endemic-infected group, $n = 12$).

Endemic participants were aged 18–25, with a median of 19 in both infected and uninfected groups. In terms of sex, 50% of the uninfected group were male, and 33% of the infected group. All participants included in this work were screened and negative for other helminth infections (*Ascaris, Trichuris, Strongyloides, Trichostrongylus, S. haematobium*, or hookworm). Participants with a history of pulmonary disease were excluded[18]. No other co-infections were assessed. Four individuals in the uninfected group had evidence for prior *S. mansoni* exposure, by antibody diagnostics (Supp. Fig. 5). Participants in the repeated CHI trial were aged 18–44, with a median of 24 in the infection control and 29 in the reinfection group. In terms of sex, 50% of the infection control group were male, and 42% of the reinfection group[16]. No CHI participants had a history of *S. mansoni* infection.

### Determination of current and prior infection
In the repeated CHI trial, volunteer infection status was determined using 500 μl of serum and the upconverting nano-particle lateral-flow CAA assay at a cut-off point ≥1 pg/ml[55]. Meanwhile, infection status in the endemic study was determined by Kato-Katz, supported by serum CAA and urine CCA (Supp. Fig. 5). Serum CAA in the endemic study was detected using 20 μl of serum and the upconverting nano-particle lateral-flow (UCP-LF CAA) assay at a cut-off point ≥10 pg/ml (the lower sensitivity reflects reduced available sample volume)[55]. Kato-Katz and urine CCA (rapid medical diagnostics) were measured at the point of sample collection[55–57]. To ascertain the prior-exposure status of Ugandan volunteers, schistosome-specific AWA-specific IgG and SEA-specific IgG were measured using in-house IFA and ELISA assays, respectively[23,58].

### PBMC isolation, storage, and thawing
PBMC isolation from endemic-infection blood samples was performed at UVRI, Uganda, while those from the repeated CHI were isolated at LUMC, Netherlands, both using the Ficoll gradient density separation method, as previously described[13].

In summary, venous whole blood samples collected in heparin tubes were diluted in HBBS (Invitrogen) and separated over a Ficoll gradient (Apotheek LUMC) by centrifugation at low break at 400×*g*, at room temperature for 25 min. PBMCs collected were then washed with HBSS, counted, and cryopreserved in complete RPMI (Invitrogen), with 100 U/ml penicillin G sodium, 100 μg/ml streptomycin (Sigma), 1 mM pyruvate (Sigma), and 2 mM glutamine (Sigma); with 10% DMSO (Merck) and 20% FCS (Bodinco). Cryopreserved cells were stored at −80° C overnight and transferred to liquid nitrogen for long-term storage. PBMC isolation and cryopreservation at UVRI followed the same protocol except for the following differences: all washes were in complete RPMI, centrifugation by Ficoll gradient at 1000×*g* for 22 min, and cryopreservation used a higher FCS percentage (50%).

Cryopreserved PBMC samples from Uganda were transported to the LUMC and thawed together with frozen PBMCs from the ReCoHSI trial. Thawing was performed at 37 °C in thawing media (complete RPMI 1640, 20% FCS, benzonase 25 units/ml (Merck).

### Stimulation for cytokine production
Up to $1 \times 10^6$ cells per condition were plated into culture media (complete RPMI and 10% FCS) in round-bottom well plates (BD Biosciences) and stimulated with culture media, SEA (10 μg/ml), CA (50 μg/ml), and AWA (50 μg/ml) for 24 h at 37 °C incubation. Frozen (eggs, cercariae) or freeze-dried (adult worms) were used to prepare antigens via homogenization with a glass homogenizer on ice in PBS, followed by sonification. Antigen preparations were frozen overnight at −80 °C, thawed, and centrifuged for 25 min at 16,200×*g* at 4 °C. Antigen preparation supernatants were filter-sterilized, and their concentration was measured via the bicinchoninic acid assay (Pierce). As a positive control, SEB (200 ng/ml) was added to pooled samples. Four hours after the beginning of incubation, Brefeldin A (5 mg/ml, Sigma) was added to wells and then incubated for a further 20 h. Upon completion of stimulation, cells were centrifuged (400×*g*, for 4 min) in a V-bottom plate (BD Biosciences) and supernatant transferred to 96-well round-bottomed plates (BD Biosciences) for subsequent cytokine analysis. Two (uninfected) endemic samples and four (infected) endemic samples were excluded due to low viability post-stimulation (<55% live) (Supp. Fig. 6). One repeated CHI participant in the infection control group was excluded from the analysis of schistosome-specific cytokines due to high cytokine production in the unstimulated control (>1%).

### Staining, flow cytometry, and analysis
Before extra- and intra-cellular staining, stimulated cells were washed twice with PBS and stained with live/dead (Blue, Thermo Fisher, 1:1000) in PBS before a third wash with FACS buffer. Extracellular staining was performed using the following antibodies; CD38 (APC-Fire, BioLegend, 1:1500, 356643), CD8 (Pacific orange, Thermo Fisher, 1:1000, MHCDO830), CD25 (BUV563, BD, 1:750, 612918), CD27(APC-H7, BD, 1:500, 560222), CD11b (BV510, BioLegend, 1:750, 101263), CD123 (BV510, BioLegend, 1:750, 306022), CD19 (BV605, BioLegend, 1:750, 302244), CD4 (cFlourYG584, CYTEK, 1:750, SKU R7-200), CD3 (BUV395, BD, 1:200,563546), CD27 (APC-H7, BD, 1:500, 560222), CD56 (BV510, BioLegend, 1:500, 318340), PD1 (BV750, BD, 1:375, 747446), γδTCR (BV480, BD, 1:250, 747446), CD127 (R718, BD, 1:100, 566967), CCR7 (BV785, BioLegend, 1:40, 353230), HLA-DR (PE-fire, BioLegend, 1:400, 307683) and human Fc block (Invitrogen, 1:200, 14-91613) to prevent non-specific antibody binding.

Intracellular staining comprised of overnight incubation with TNF (PE-Cy7, BD, 1:3000, 557647), IL-17A (Pacific blue, BioLegend, 1:3000, 512312), IL-5 (APC, BioLegend, 1:1000, 504306), IL-4 (BUV737, BD, 1:750, 612835), IFN-γ (BV650, BD, 1:750, 563416), IL-13 (BV711, BD, 1:750, 564288), IL-9 (PE, BioLegend, 1:500, 507605), IL-10 (PerCP ef710, Thermo Fisher, 1:250, 46-7108-4), CTLA (PE-Cy5, BD, 1: 10000, 555854),

IL-21 (AF647, BD, 1:200, 560493), FoxP3 (PE-Dazzle594, BioLegend, 1:150, 320126), IL-22 (Vio515, Miltenyi, 1:100, 130-108-09), GATA3 (BV421, BD, 1:200, 563349), CD45RA (BUV496, BD, 1:750, 741182), CD45RA (BUV496, BD, 1:750, 741182), CD45RO (BUV805, BD, 1:750, 748367) and CRTh2 (BUV661, BD, 1:200, 741663) antibody mix diluted in eBio perm buffer (eBioscience). Stained cells were washed thrice in eBio perm buffer (200 μl, eBioscience) and acquired on a spectral flow cytometer (Cytek Aurora). Cytometer output was analyzed in Spectral flow version 3.1 (Cytek) and OMIQ (Dotmatics) software, and gates were positioned using a combination of fluorescence minus many (FMM) controls and medium as negative controls.

## Measurement of serum cytokines
The following cytokines: CCL4, CCL23, CXCL10, IL-5, IL-13, IL-10, IL-18, TNF, and IFN-γ were measured in serum and cell supernatant using a commercial Luminex kit (Cat. Number: LXSAHM, R&D Systems), according to the manufacturer's instructions. Cytokines for which over 60% of samples were below the limit of detection were excluded from analysis. For serum cytokines, these were IL-5, IL-13, and IFN-γ, and IL-5, CCL23, and IFN-γ IL-18 for supernatant cytokines.

## Statistical analysis
Statistical analysis and visualization was done in R (v4.3.1 core team 2023) and R Studio (version 2023.06.1)[59]. Response to repeated CHI infection-control and reinfection groups over time was assessed using a linear mixed model (R package lmerTest v3.1–3)[60] with timepoint as a factor and fixed effect, and participant as a random effect. The frequency of T cell subsets and cytokine-positive CD4 + T cells were used as response variables during analysis to determine T cell and cytokine response outcomes, respectively. P values were FDR-corrected for multiple comparisons. To compare reinfection (all) and infection-control groups, Welch's T-tests was performed at each time point. Normality was not assumed for repeated CHI m-f-m or any endemic groups. Therefore, in comparing between reinfection (m-m-m) and reinfection (m-f-m) groups Mann–Whitney tests were performed at each timepoint. To compare immune responses between individual reinfection volunteers at weeks 26 and 30 Wilcoxon signed-rank test was used (package rstatix v 0.7.2)[61]. To compare between endemic infected and uninfected, a Mann–Whitney test was used, with a Kruskal–Wallis test, followed by post hoc Dunn's test (package rstatix v 0.7.2)[61] to compare to reinfection groups. For all cross-group comparisons, the r package rstatix was used. All statistical tests were two-tailed. Tables with all statistical comparisons, including exact p values are included in the Source Data File.

## Inclusion and ethics
Local researchers (from the endemic region) have been included throughout the research process. This is reflected in authorships as well as citations included in the article. This research has been approved in the endemic region (Uganda) as well as the non-endemic region (Netherlands).

## Reporting summary
Further information on research design is available in the Nature Portfolio Reporting Summary linked to this article.

# Data availability
All the data generated or analyzed during this study are included in this paper, its Supplementary Information, and the Source Data file. This paper does not include any data type with mandatory deposition. Source data are provided with this paper.

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

## Acknowledgements

The authors thank the LUMC Klinisch Microbiologisch Laboratorium (KML) for performing antibody diagnostics in this work. Most of all, we thank all the study volunteers for their participation. This research has received funding from the European Union under ERC St grant agreement No 101075876. Part of the work was conducted at the MRC/UVRI and LSHTM Uganda Research Unit, which is jointly funded by the UK Medical Research Council (MRC), part of UK Research and Innovation (UKRI), and the UK Foreign, Commonwealth and Development Office (FCDO) under the MRC/FCDO Concordat agreement and is also part of the EDCTP2 programme supported by the European Union. Harriet Mpairwe received a Wellcome Training fellowship 102512, ELH was supported by the European Union's Horizon 2020 research and innovation program under the Marie Skłodowska-Curie grant (101063914), and ED was supported by a Wellcome grant for "Human infection studies for *Schistosoma mansoni* vaccine testing in Uganda" (218454z/19/z). The work was also supported by funding from the European Union [Project 101080784—WORMVACS2.0]. Funding sources did not play any role in the data collection, analysis, interpretation, and or data reporting. Opinions expressed in this paper reflect only the authors' views, and the funding sources are not responsible for the use that maybe made of and from the contents of this document.

## Author contributions

E.L.H., E.D., M.Y., M.R., J.P.R.K., H.M., A.S.M., M.E., and A.M.E. were responsible for study conceptualization and design. E.D., K.A.S., F.S., and E.L.H. were responsible for the statistical analysis and data interpretation and prepared the first draft. M.Y., M.R., A.M.E., H.M., and A.S.M. acquired funding. J.P.R.K. and M.R. were the clinical investigators. J.J.J. performed the trial management. M.C-P., J.S., and J.J.J. were responsible for the production and release of cercariae. F.S., M.K., Y.K., S.T.H., S.S., L.d.B-R., E.I., E.D., and E.L.H. generated the data and optimized the experimental protocols. L.d.B-R., E.I., I.N., M.K., and Y.K. coordinated PBMC sample collection and processing. S.T.H., G.J.v.D., P.L. A.M.C., and L.v.L. supported diagnostic (CAA, IFA, and ELISA) tests. All authors contributed to the manuscript review.

## Competing interests

The authors declare no competing interests.
