## [Peer Review file · Nature Communications]

T cell responses in repeated controlled human schistosome infection compared to natural exposure

Corresponding Author: Dr Emma Houlder

Version 0:

Reviewer comments:

Reviewer #1

(Remarks to the Author)

OVERALL ASSESSMENT

In this study, Driciru et al. provide a comprehensive analysis of T cell and cytokine responses in *Schistosoma mansoni* reinfection and treatment cycles, employing a controlled human infection (CHI) model in comparison to schistosome-endemic infections. The authors utilize advanced cytometry techniques, which add valuable insights into the understanding of immune responses. The comparison between endemic and non-endemic populations strengthens the quality of the study. However, several aspects require clarification and improvement to enhance the manuscript's impact and interpretability.

MAJOR COMMENTS

1. The manuscript would benefit from a clearer articulation of the primary hypothesis and specific objectives in the introduction. While the study investigates immune responses during reinfection, it is not explicitly stated whether the aim is to explore protective immunity, immune modulation, or both. A more precise statement of intent would help contextualize the findings.
2. The authors emphasize that their experiments highlight the translatability of the CHI model to natural infection in endemic areas. However, this claim may be premature. While their findings suggest similarities in T cell phenotypes and serum cytokine levels, notable differences and limitations are also observed but not fully discussed. A more balanced interpretation is necessary, acknowledging both the strengths and constraints of the CHI model.
3. The manuscript discusses immune differences between CHI volunteers and endemic individuals, attributing them to environmental factors. However, a more detailed characterization of the endemic population would strengthen this argument. Key demographic and epidemiological variables—such as age, history of prior infections, and co-infections—should be included to rule out potential confounding factors.
4. The authors attribute differences in immune responses within male-female-male group to egg production. However, given that only one out of five participants had detectable eggs in stool, this conclusion is not supported by their findings. Could the observed differences instead be driven by proteins exclusively produced by female worms? Further discussion or additional data would help clarify this point.

MINOR COMMENTS

- Some statements in the discussion are speculative and should either be supported by additional references or softened in tone.
- Figure legends should be more detailed, particularly in explaining abbreviations. For example, in Figure 1B, what does "CM" stand for? It is likely "Central Memory," but this is not explicitly stated in the text or figure legend.
- There is no reference to Figure 1E in the text.
-

CONCLUSION

This study provides important insights into immune responses to *S. mansoni* infection, offering valuable contributions to the field. However, refining the clarity of the hypotheses and addressing concerns mentioned above — particularly regarding the discussion — would significantly enhance the manuscript's impact. Addressing these aspects will improve the robustness of the conclusions.

RECOMMENDATION: MINOR TO MODERATE REVISIONS

This article requires minor revisions and is undoubtedly of considerable interest to researchers in this field, with the potential

to inspire new perspectives. However, I am not fully convinced that its findings represent a sufficiently significant advancement to justify publication in Nature Communications, as the results appear to be a continuation of a previous study published by the same research group.

Reviewer #2

(Remarks to the Author)

This report presents a comprehensive analysis of T cell immune response development in the controlled human infection (CHI) model for schistosomiasis. It is very informative and useful to increase understanding of the human host immune response to schistosome infection. Perhaps the most interesting aspect of the paper is a result of the mistake of exposing some of the participants to female cercariae and observing their putative first exposure to eggs alters their responses compared to participants that only received male cercariae or individuals with chronic infection from an endemic area. However, because this was a protocol deviation (albeit unintentional), it is unlikely that this aspect of the study can be replicated. But it is still worthwhile to report the results here.

The only real complaint about the paper is the last sentence of the discussion. It is not clear from what is written as to what the value of transferring the CHI model to an endemic area will be. The all-male cercarial exposure immune responses differ little from the endemic infected individuals (except for maybe IL-17 responses--yes, they are statistically different, but the changes are in the hundredths of a percent--biologically relevant?). It is conceivable that the lack of egg exposure in the all-male group reduces the magnitude of the response (e.g., vs. m-f-m exposed individuals) and the chronically infected persons are so immunoregulated that those responses are dampened (see IL-10:TNF ratios in Fig 4). If there are no differences in the Dutch and Ugandan populations, the results of m-m-m exposure and chronic endemic infections would likely be similar. It is not clear that establishing that Dutch CHI responses are the same as Ugandan CHI responses would be instructive enough to merit the complications associated with establishing the CHI model in Uganda. If there is a good reason to do this, the argument needs to be made more clearly.

Some minor comments and one quibble:

--in the Study design section of the methods, a comment is made that treatments were made with praziquantel or placebo, but there is no other mention of placebo in the paper--perhaps its inclusion here was an oversight?

--in the ribbon plots, asterisks are either black or the color of the line representing the infection group. The black asterisks show differences between the curves at different time points, but it is not quite clear what the colored asterisks represent, presumably it is differences at that time point to the baseline values of that group? This just needs direct clarification in the legend. For the comparison of the curves, would repeated measures ANOVA be a more appropriate statistical test as it shows the groups are different over time rather than different at one time point and not another?

--in 5th paragraph of the discussion, the authors state that the dampened egg response may develop as a protective mechanism against egg-induced host tissue damage. The quibble that chronic pathology is not caused by the eggs themselves but the fibrogenic immune response to them. The authors clearly know this, the sentence just needs rewording.

Dear Reviewers,

Thank you for allowing us to submit a revised draft of our manuscript for publication. We are grateful for the time and effort you have put into assessing the manuscript and providing insightful feedback. Changes to the manuscript as a result of your comments are shown as tracked changes in the marked version. Please see below our point-by-point answers (in blue) to your concerns. Line numbers are accurate in the unmarked manuscript (“Manuscript – changes unmarked”). We believe the comments have substantially improved the manuscript.

REVIEWER COMMENTS

Reviewer #1 (Remarks to the Author):

OVERALL ASSESSMENT

In this study, Driciru et al. provide a comprehensive analysis of T cell and cytokine responses in *Schistosoma mansoni* reinfection and treatment cycles, employing a controlled human infection (CHI) model in comparison to schistosome-endemic infections. The authors utilize advanced cytometry techniques, which add valuable insights into the understanding of immune responses. The comparison between endemic and non-endemic populations strengthens the quality of the study. However, several aspects require clarification and improvement to enhance the manuscript’s impact and interpretability.

MAJOR COMMENTS

1. The manuscript would benefit from a clearer articulation of the primary hypothesis and specific objectives in the introduction. While the study investigates immune responses during reinfection, it is not explicitly stated whether the aim is to explore protective immunity, immune modulation, or both. A more precise statement of intent would help contextualize the findings.

Response

Thank you for your comments. We appreciate that the precise objectives of this exploratory study could be better elucidated. The previously published clinical arm of this trial investigated whether reinfection could induce protective immunity in the repeat CHI volunteers, and found that reinfection does not result in protective immunity although a decrease in symptoms was observed (lines 65-67). The precise objectives of our study have now been added to the introduction section (lines 77-84):

“In this exploratory immunological study, we investigated the specific cellular and cytokine response overtime during repeat CHI and compared this to endemic natural infection. We aimed to, first, determine the cellular, and CD4+ T cell-specific cytokine response to Sm adult-, cercarial- and egg- antigens during repeat CHI. Secondly, to compare the Sm-specific T cell

profile, and CD4+ T cell cytokine response to Schistosoma life-stage specific antigens during repeat CHI to that in endemic natural infection. We hypothesized that experimental Schistosoma reinfection exerts an immune modulatory response in the human host similar to that in natural endemic settings, with possible immune-protective impact.”

2. The authors emphasize that their experiments highlight the translatability of the CHI model to natural infection in endemic areas. However, this claim may be premature. While their findings suggest similarities in T cell phenotypes and serum cytokine levels, notable differences and limitations are also observed but not fully discussed. A more balanced interpretation is necessary, acknowledging both the strengths and constraints of the CHI model.

Response

Thank you for your feedback, which we believe has greatly improved the quality of the discussion. In response we now explicitly addressed the strengths and limitations of this model.

Limitations (discussion section, lines 313-320):

“Moreover, CHI models are limited by their incapacity to replicate the complex natural immunological and environmental context. Challenge strains often do not represent circulating wild-types, and CHI volunteers such as in this repeat CHI, may differ from vaccine target populations in terms of comorbidities, microbiome, immunogenetics and prior-exposure history (56). A limitation of this repeat CHI modes is that the initial design was intended for a single-sex male-only reinfection. Hence, whether male-female worm interaction (and egg-production) occurred in the unintended m-f-m group remains uncertain given that stool PCR was negative for 4/5 participants.”

Strengths (discussion section, lines 322-328):

“Despite these limitations, the CHI model provides a precise alternative to natural infection studies where heterogenous pre-exposure levels, undefined infection timing and inconsistent infectious doses limit definitive investigations into infection and disease prognosis. The controlled standardized infection conditions in CHI allows for detailed longitudinal studies of pathogen-host interactions, with a small number of volunteers (56). Using this system we have demonstrated how immune responses can develop during repeated S. mansoni infection/treatment cycles. Beyond progressing scientific understanding, CHI models can provide proof-of-concept for vaccine efficacy (57).”

In addition to this, we have now tempered our claims regarding translatability, emphasizing that while our findings suggest shared immunological features, differences do exist between endemic and repeat CHI volunteers (Discussion section, lines 332-335).

“However, immunological differences still exist between the two groups, as demonstrated by the non-overlapping PCA clustering, including increased IL-17 levels and reduced Th1 (IFN- γ and TNF) responses in endemic infected volunteers compared to repeated m-m-m repeated CHI. “

3. The manuscript discusses immune differences between CHI volunteers and endemic individuals, attributing them to environmental factors. However, a more detailed characterization of the endemic population would strengthen this argument. Key demographic

and epidemiological variables—such as age, history of prior infections, and co-infections—should be included to rule out potential confounding factors.

Response

Thank you for your feedback. In response we have now added further demographic information into the methods (lines 363-371):

“Endemic participants were aged 18-25, with a median of 19 in both infected and uninfected groups. In terms of sex, 50% of the uninfected group were male, and 33% of the infected group. All participants included in this work screened and negative for other helminth infections (Ascaris, Trichuris, Strongyloides, Trichostrongylus, S. haematobium, or Hookworm). Participants with a history of pulmonary disease were excluded (58). No other co-infections were assessed. Four individuals in the uninfected group had evidence for prior S. mansoni exposure, by antibody diagnostics (Supp. Figure 5). Participants in the repeated CHI trial were aged 18-44, with a median of 24 in the infection control and 29 in the reinfection group. In terms of sex, 50% of the infection control group were male, and 42% of the reinfection group (17). No CHI participants had a history of S. mansoni infection. “

In addition, we have now discussed the demographic differences between the two populations (lines 299-304):

“A limitation of this study is that we could not comprehensively screen participants for prior and current (co-)infections, with exposure histories likely differing in endemic and CHI volunteers, potentially contributing to observed differences. For example, over 95% of 5 year old children in the endemic area of Entebbe, Uganda has experienced cytomegalovirus, and 60% malaria infection by age 5 (53) (Ref). Malaria is not present in the Netherlands, and the lifetime seroprevalence of cytomegalovirus in the Netherlands is only 45% (54).”

4. The authors attribute differences in immune responses within male-female-male group to egg production. However, given that only one out of five participants had detectable eggs in stool, this conclusion is not supported by their findings. Could the observed differences instead be driven by proteins exclusively produced by female worms? Further discussion or additional data would help clarify this point.

Response

We acknowledge that the observed immune responses in the m-f-m could be attributed to factors beyond detectable egg output such as the presence of female worm-derived antigens. As such, we have revised the manuscript’s discussion section to include a paragraph exploring possibility of female worm secreted proteins and alternative explanations underlying the observed immune modulation, citing relevant literature (lines 275-285):

“However, egg presence was not confirmed by egg microscopy or PCR in most (4/5) of these volunteers, potentially due to insufficient sensitivity to detect low-levels of eggs. The observed response may stem from factors beyond detectable egg output, notably, response to female worm-specific antigens. Murine studies have shown differential immune responses in female compared to male infection, and female worms express distinct proteins and glycosylation patterns which may modify host immune response even in absence of patent egg production (24, 38-41). However, in our previous single-sex CHI models, immune profile and magnitude in the female-only was comparable to that in the male-only CHI model (14), suggesting female-specific factors alone may not sufficiently explain the m-f-m reinfection response. Alternatively,

proteins expressed during pairing, mating and or early stages of oogenesis following potential male-female worm interaction could underlie the observed response (24, 40, 42). ”

MINOR COMMENTS

- Some statements in the discussion are speculative and should either be supported by additional references or softened in tone.

Response

Thank you, in response to this comment and those by the other reviewers we have revised the manuscript to provide a more nuanced interpretation of our findings in the discussion section.

- Figure legends should be more detailed, particularly in explaining abbreviations. For example, in Figure 1B, what does "CM" stand for? It is likely "Central Memory," but this is not explicitly stated in the text or figure legend.

Response

Thank you for your feedback. In response, we have added more detail to the figure legends, including explanations of abbreviations in both the figures and the legends.

- There is no reference to Figure 1E in the text.

Response

Reference to Figure 1E has been inserted next to the sentence that describes it in the results section of the manuscript text (line 111). It now reads;

“Similarly, at week 9 and 26 a CD38⁺CD27⁺ DN T cells increased significantly in the reinfection (all) group, as well as week 26 in the infection control group (Fig. 1E)”. (Further discussion on Fig. 1E can be found in the second paragraph of the discussion section; lines 244-251).

CONCLUSION

This study provides important insights into immune responses to *S. mansoni* infection, offering valuable contributions to the field. However, refining the clarity of the hypotheses and addressing concerns mentioned above — particularly regarding the discussion — would significantly enhance the manuscript’s impact. Addressing these aspects will improve the robustness of the conclusions.

RECOMMENDATION: MINOR TO MODERATE REVISIONS

This article requires minor revisions and is undoubtedly of considerable interest to researchers in this field, with the potential to inspire new perspectives. However, I am not fully convinced that its findings represent a sufficiently significant advancement to justify publication in Nature Communications, as the results appear to be a continuation of a previous study published by the same research group.

Reviewer #2 (Remarks to the Author):

This report presents a comprehensive analysis of T cell immune response development in the controlled human infection (CHI) model for schistosomiasis. It is very informative and useful to increase understanding of the human host immune response to schistosome infection. Perhaps the most interesting aspect of the paper is a result of the mistake of exposing some of the participants to female

cercariae and observing their putative first exposure to eggs alters their responses compared to participants that only received male cercariae or individuals with chronic infection from an endemic area. However, because this was a protocol deviation (albeit unintentional), it is unlikely that this aspect of the study can be replicated. But it is still worthwhile to report the results here.

The only real complaint about the paper is the last sentence of the discussion. It is not clear from what is written as to what the value of transferring the CHI model to an endemic area will be. The all-male cercarial exposure immune responses differ little from the endemic infected individuals (except for maybe IL-17 responses--yes, they are statistically different, but the changes are in the hundredths of a percent--biologically relevant?). It is conceivable that the lack of egg exposure in the all-male group reduces the magnitude of the response (e.g., vs. m-f-m exposed individuals) and the chronically infected persons are so immunoregulated that those responses are dampened (see IL-10: TNF ratios in Fig 4). If there are no differences in the Dutch and Ugandan populations, the results of m-m-m exposure and chronic endemic infections would likely be similar. It is not clear that establishing that Dutch CHI responses are the same as Ugandan CHI responses would be instructive enough to merit the complications associated with establishing the CHI model in Uganda. If there is a good reason to do this, the argument needs to be made more clearly.

Response

We appreciate the reviewer's thoughtful critique regarding similarity between male-only and endemics, and subsequently the value of transferring the model to an endemic setting.

Differences between the m-m-m repeat CHI and natural endemic infection groups are now discussed further (lines 333-336).

"immunological differences remain between the two groups, as demonstrated by non-overlapping PCA clustering, including increased IL-17 levels and reduced Th1 (IFN- γ and TNF) responses in endemic infected volunteers compared to repeated m-m-m CHI".

Therefore, transferring this model in an endemic setting would provide further insights into the observed immune dynamics using the controlled longitudinal design of CHI while taking into account how prior-exposure, repeated-praziquantel-treatment, coinfections, and environmental factors may shape immune responses in clinically relevant ways. The last sentence of the discussion section now incorporates the rationale for a transfer to an endemic setting and reads as;

"Therefore, transferring the CHI model to an endemic area is a critical next step to gain a holistic insight into how prior-exposure, co-infections, immune regulation and repeated treatment may influence host responses to Sm (re)infection and vaccine-candidates (45) using the controlled longitudinal CHI design." (Last paragraph of the discussion section; lines 336-339).

Some minor comments and one quibble:

--in the Study design section of the methods, a comment is made that treatments were made with praziquantel or placebo, but there is no other mention of placebo in the paper--perhaps its inclusion here was an oversight?

Response

Thank you for the opportunity to clarify this. *"At weeks 8 and 17, the infection control group was given a placebo treatment in place of praziquantel."* This has now been added to the methods section (line 357).

--in the ribbon plots, asterisks are either black or the color of the line representing the infection group. The black asterisks show differences between the curves at different time points, but it is not quite clear what the colored asterisks represent, presumably it is differences at that time point to the baseline values of that group? This just needs direct clarification in the legend. For the comparison of the curves, would repeated measures ANOVA be a more appropriate statistical test as it shows the groups are different over time rather than different at one time point and not another?

Response

Thank you for this comment. We have now added further annotation to the figures to clarify what the asterisks represent. In response to the statistics query, we considered using a general test such as repeated measures ANOVA to assess whether the groups differ over time. However, since differences between groups occur only at specific timepoints, a global interaction effect may not capture these localized changes. We therefore chose an approach that allows for detecting timepoint-specific differences.

--in 5th paragraph of the discussion, the authors state that the dampened egg response may develop as a protective mechanism against egg-induced host tissue damage. The quibble that chronic pathology is not caused by the eggs themselves but the fibrogenic immune response to them. The authors clearly know this, the sentence just needs rewording.

Response

We agree that the original phrasing "dampened egg response may develop as a protective mechanism against egg-induced host tissue damage" suggests that the eggs themselves directly cause chronic pathology. In effect to this comment, we have revised the sentence to emphasize that it is the immune response to the eggs that drives tissue damage. The revised sentence now reads;

"This dampened egg response may serve to limit immunopathology arising from the host's fibrogenic reaction to deposited eggs." (line 295)